# Ultrahigh Extinction Ratio Leaky-Guided Hollow Core Fiber Mach–Zehnder Interferometer Assisted by a Large Core Hollow Fiber Beam Splitter

**DOI:** 10.3390/nano14181494

**Published:** 2024-09-14

**Authors:** Yan-Han Lu, Ren-Xiang Luo, Cheng-Ling Lee

**Affiliations:** Department of Electro-Optical Engineering, National United University, Miaoli 360, Taiwan; a131085258@gmail.com (Y.-H.L.); a0981262950@gmail.com (R.-X.L.)

**Keywords:** fiber Mach–Zehnder interferometer (FMZI), extinction ratio (ER), hollow core fiber (HCF), leaky-guided fiber waveguide (LGFW), free spectral range (FSR), interference

## Abstract

We proposed a novel fiber Mach–Zehnder interferometer (FMZI) that can perform an ultrahigh extinction ratio (ER), ultracompact, and ultra-broadband interference characteristics. The FMZI structure is based on an extremely tiny hollow core fiber (HCF) with a small diameter of 10 μm (named HCF_10_) connected with a beam splitter of a large core of 50 μm HCF (named HCF_50_). The refractive index (RI) of the air core is lower than that of the HCF cladding; a leaky-guided fiber waveguide (LGFW) occurs in such a short-section HCF_10_ waveguide to simultaneously have the core and cladding modes. To achieve better fringe visibility of the interference, the section of HCF_50_ assists in splitting the optical light into core and cladding beams launched into the HCF_10_ with appropriate intensities. Experimental and simulation results show that the optical characteristics of the proposed LGFW-FMZI are very similar. Based on the results of the study, the length of the HCF_10_ primarily influences the free spectral range (FSR) of the interference spectra, and the HCF_50_ splitter significantly controls the optimal extinction ratio (ER) of the interference fringes. By exactly adjusting the lengths of HCF_10_ and HCF_50_, the proposed fiber interferometers can perform the capability of an ultrahigh ER over 50 dB with the arbitrary FSR in the transmitted interference spectra over an ultra-broad wavelength range of 1250 nm to 1650 nm.

## 1. Introduction

In recent years, fiber optic sensing technology has become increasingly mature. Due to the advantages of optical fibers, such as high bandwidth, low loss, and resistance to electromagnetic interference, many research topics focused on fiber-optic sensors/devices have gradually gained attention. Among these advanced fiber optic sensors/devices, in-line all-fiber interferometers are well-known for measuring interference signals due to their high sensitivity. Their mechanism, which causes significant changes in optical path difference and shifts interference fringes with minimal variations in the measured parameters, is highly advantageous and sensitive. Additionally, the all-fiber light-guiding mechanism, which allows for real-time signal monitoring without the need for light alignment, has attracted significant interest. However, there are numerous types of all-fiber interferometers, each with advantages and disadvantages. Among them, the structure based on hollow-core fibers (HCF), exceptionally pure silica hollow capillaries, and easy fusion splicing with commercial single-mode fiber (SMF) is renowned for its straightforward structure and the ability to fill various analytes or other materials into the fiber core for modulating optical waveguide characteristics [1,2,3,4,5,6,7,8,9,10,11,12,13,14,15,16,17,18,19,20,21,22]. Therefore, in this kind of HCF-based fiber interferometer structure, the materials are not contaminated by external factors and are protected by the HCF, ensuring stability and preventing deterioration, which is a significant advantage. Moreover, the materials with the minimal volume required (typically on the picoliter scale) and high sensitivity are additional driving forces for many research groups worldwide. For example, a fiber device was based on a liquid-core optical fiber by filling the nonlinear materials of carbon disulfide (CS_2_) into an HCF to achieve an ultralow threshold Raman generation [1]. A new surface plasmon resonance (SPR) sensor is presented based on a silver-coated inside the HCF, and a liquid-sensed medium with high RI is filled in the hollow core, and its refractive index (RI) can be detected by measuring the transmission spectra of the HCF-based sensor [2]. In 2021, Zhang et al. proposed an electrically tunable optical fiber device based on the fiber Mach–Zehnder interferometer (FMZI) and the electro-optic effect of the liquid crystal. The electric field modulates the resonance wavelength of the spectrum transmitted from the proposed HCF-based FMZI sensor, which has potential applications in wavelength-tunable electro-optical devices, photoelectric switches, and electric field sensors [3]. Thus, numerous in-line advanced applications of the HCF-based interferometers, including the fiber Fabry–Perot interferometer (FFPI) [4,5,6,7,22], fiber Mach–Zehnder interferometers (FMZI) [8,9,10,11,12,13,14,15,16,17], and anti-resonance (AR) fiber interferometer (ARFI) [18,19,20,21], were published constantly. The mentioned work of the above FFFI with HCF-filled polymers, air, and optical liquids has been reported for sensing the airflow [4], temperature [5,6], RI [7], and thermal optics coefficient (TOC) [22], respectively. Compared to FFPIs, the FMZIs seem more attractive and beneficial because the latter always operate in the transmitted fiber mode for measuring the optical responses by a detector or an optical spectra analyzer (OSA). The principle of any FMZI involves splitting incident light into two beams (usually core and cladding modes), which travel different paths and then recombine, creating interference from the optical phase difference due to the optical path length difference in the two modes. In 2015, an FMZI interferometer was formed by tapering HCF to create beam splitting/combining for the two beams interference [8]. Another study used hygroscopic materials coated on the HCF to form the FMZI for humidity sensing [9]. An HCF-based FMZI structure with double abrupt tapers for simultaneous temperature and bending measurement was proposed [10]. S. Liu et al. proposed a liquid filled into HCF to modify the waveguide characteristics of the core and cladding core to generate the liquid core FMZI for simultaneous measurement of RI and temperature [11]. A stress sensing using the tapered HCF formed the FMZI has been reported [12]. In 2020, a curvature fiber sensor based on the connection of the SMF-MMF-HCF-MMF-SMF structure was presented [13]. The above HCF-based FMZIs have useful applications but would not be very compact [2,8,9,10,11,12,13]. Another type, the SMF-HCF-SMF structure of ARFI, is based on the multiple anti-reflection beam interference introduced by the cladding of the HCF. They can generate periodic interference dips with a high extinction ratio (ER). However, the lengths of the used HCF with millimeters and centimeters are required [14,15,16,17,18,19]. To achieve the purpose of miniaturizing the HCF-based fiber interferometers, an ultra-compact in-line broadband Mach–Zehnder interferometer using a composite leaky hollow-optical fiber waveguide was the first presented by K. Oh et al. [20]. The study used a hundred-micrometer-length HCF to generate leaky-guided air core mode and cladding mode interference in the FMZI scheme. In 2022, we proposed a novel sensor based on an ultra-compact leaky-guided liquid core FMZI for measuring the liquids’ RI and TOC. The sensor structure is based on a micro-sized HCF splicing a tilted end-face SMF to create a miniature oblique gap for liquid filling [21]. Although a miniature structure of the above HCF-based fiber interferometers was achieved, a high fringe visibility (i.e., high ER) performance is essential and crucial in the optical interference spectra.

In this study, we propose an ultrahigh ER and ultracompact leaky-guided FMZI based on connections of various diameters and micro-lengths of HCFs to create an SMF_in_-HCF_50_-HCF_10_-SMF_out_ structure. The core diameters of the hollow core for the HCF_50_ and HCF_10_ are 50 μm and 10 μm, respectively. By appropriately varying the micro-lengths of the two HCF_50_ (L_50_) and HCF_10_ (L_10_) sections, we demonstrate the light from input SMF_in_ can be split by the HCF_50_ and generate two paths in the HCF_10_, then combining two paths in output SMF_out_ to optimize the optical interference spectra. Both experimental and simulated results indicate that the free spectral range (FSR) of the interference spectra in the FMZI is primarily influenced by the interference section L_10_ of the HCF_10_. However, the ER of the interference spectra would be greatly affected by the L_50_ of beam splitter HCF_50_ when the L_10_ is fixed. The simulation results using a numerical finite difference beam propagation method (FDBPM) show that the optimal performance of ultrahigh ER over 50 dB can be achieved using the various structures of the combination of L_50_ and L_10_. Experimental results show that the HCF_50_ has significant assistance from light splitting to achieve excellent interference, with the highest over 40 dB, primarily consistent with the simulation results.

## 2. Principle of the Proposed Leaky-Guided FMZI

In this study, a head input SMF_in_ was first fusion spliced to a length of L_50_ of HCF_50_ and, at the end of the HCF_50_, successively splicing another section HCF_10_ with a length of L_10_. The inner air-core diameters of the HCF_50_ and HCF_10_ are 50 μm and 10 μm, respectively, and their outer silica-cladding is 125 μm. Finally, an output SMF_out_ collects the optical signal to be measured. Figure 1a schematic plots the configuration of the proposed leaky-guided FMZI splits the leaky-core mode and cladding modes for the interference. To understand more about the section of the HCF_50_ and how it assists in splitting the optical light into two paths of the leaky core and cladding beams that launch into the HCF_10_ with appropriate intensities. The FDTD method of the RSoft BeamPROP tool (Synopsys, Inc., located in Mountain View, CA, USA) is useful for theoretically calculating and analyzing optical light characteristics in photonic structures. The used simulation tool, a scalar beam propagation method (BPM) in the RSoft suite, is suitable for calculating light propagation along the *z*-axis in complicated fiber structures and optical waveguides. In the calculation, the core and cladding diameters of the SMF are set to be 8.2 μm and 125 μm, respectively, and their RIs of the core $n_{\mathrm{co}\;}\left(\lambda \right)\;$ and cladding $n_{cl}\;(\lambda )$ are based on the Sellmeier equations of the dispersion, as shown below [23].
(1)$$n_{\mathrm{co}\;}\left(\lambda \right)=\sqrt{1+\frac{0.700071\times \lambda^{2}}{\lambda^{2}-0.004596176}+\frac{0.421598\times \lambda^{2}}{\lambda^{2}-0.01448869}+\frac{0.888017\times \lambda^{2}}{\lambda^{2}-98.203859}}\;(\lambda \;in\;\mu m)$$
(2)$$n_{\mathrm{cl}}\;\left(\lambda \right)=\sqrt{1+\frac{0.6961663\lambda^{2}}{\lambda^{2}-0.004679148}+\frac{0.4079426\lambda^{2}}{\lambda^{2}-0.01351206}+\frac{0.8974794\lambda^{2}}{\lambda^{2}-97.934002}}\;(\lambda \;in\;\mu m)$$

Figure 1b shows the BPM simulation results of the optical field distribution of light propagating at fiber communication wavelength (λ) of 1550 nm along the proposed successive segment of HCFs waveguide when the hollow core is set to be air (n = 1). Figure 1b shows that the optical mode field expands in section HCF_50_ and splits into two paths, launching into the HCF_10_ section. In the HCF_10_ segment, the RI of its core is lower than that of the cladding, and a leaky light appears in this section to simultaneously have the core and cladding modes for obtaining interference to the SMF_out_. From the optical mode field distribution of light in Figure 1b, the superposition of the optical waves produces constructive and destructive optical fields. On the contrary, Figure 1c plots the optical field distribution of light without the segment of the HCF_50_ beam splitter. The leaky light also appears in the HCF_10_ section to generate core and cladding paths; however, much of the optical intensity is concentrated in the core of the tiny HCF_10,_ not to achieve excellent interferences. Therefore, we believe that by appropriately arranging the lengths of HCF_50_, HCF_10_*,* and RI of the core, the results can obtain excellent interference characteristics with ultrahigh ER and arbitrary FSR in the optical interference spectra of the proposed LGFW-FMZI.

In the studied leaky-guided FMZI, the optical intensity of cladding and core modes are denoted as I_cl_ and I_co_, respectively. The total intensity of the interference beam is denoted as I_FMZI_, defined by the following Equation (3) [21].
(3)$$I_{FMZI}=I_{co}+I_{cl}+2\sqrt{I_{co}\cdot I_{cl}}cos(\frac{2\pi }{\lambda }\Delta n^{eff}L_{10})$$
where L_10_ is the length of HCF_10,_ which is the interference length, and λ is the wavelength of the optical light. $\Delta n^{eff}=n_{cl}^{eff}-n_{co}^{eff}$ is the effective index difference and $n_{cl}^{eff}$ and $n_{co}^{eff}$ present the effective refractive index of cladding and core modes generating the interference, respectively. When the optical phase difference ((2π/λ)·Δn^eff^·L_10_) matches the condition of destructive interference, the dip wavelength ($\lambda_{min}^{m}$) of minimum power can be deduced as Equation (4) below.
(4)$$\lambda_{min}^{m}=\frac{2}{2m+1}\Delta n^{eff}\cdot L_{10}$$

Here, m is the order of interferential mode, and it is an integer. The FSR of the FMZI means the wavelength difference in two continuous interference dips or peaks between the front and rear. Afterwards, we can utilize the relationship of interference phase difference to derive FSR, which can be expressed as Equation (5).
(5)$$FSR=\left|\lambda_{1}-\lambda_{2}\right|=\frac{\lambda_{1}\lambda_{2}}{\left|n_{cl}^{eff}-n_{co}^{eff}\right|L_{10}}$$
where λ_1_ and λ_2_ represent the wavelengths of two adjacent interference dips. To achieve optimal interference, the contrast of the interference fringes should be maximized, meaning the amplitude of the interference term should be at its maximum. In Equation (3), the amplitude of the interference term $2\sqrt{I_{\mathrm{co}\;}{\cdot I}_{\mathrm{cl}\;}}\;$that reaches its maximum value when the $I_{\mathrm{co}\;}approaches\;I_{\mathrm{cl}\;}$. Thus, for the best interference fringe visibility, the intensities of the core mode and the cladding mode should be very close.

## 3. Simulation and Experimental Results

In the simulation, we arbitrarily adjust the lengths of HCF_50_ (L_50_) and HCF_10_ (L_10_) to vary the core mode (I_co_) and cladding mode light intensity (I_cl_) for the analysis of the interference spectra in the proposed successive segment HCF structure. To achieve the optimal interference results, we adjusted the lengths of segments HCF_50_ and HCF_10_ to make the I_co_ approach the I_cl_. As the structure is shown in Figure 1a, a Gaussian power light source was set with a wavelength range from λ = 1250 nm to 1650 nm (resolution: 0.8 nm) that is launched from the input SMF_in_. Here, all the refractive indices of the used fiber core and silica cladding are considered chromatic dispersion over such a wide wavelength range over 400 nm. Figure 2 illustrates the simulation results of interference spectra for the different length combinations of the HCF_50_ and the HCF_10_ in the leaky-guided FMZI structure. Here, the core of HCF is air (n = 1). The optimal interference patterns are obtained by varying the length L_50_ from 50 to 140 μm with an interval of 5 μm when fixing the length L_10_ of HCF_10_. The results are shown in Figure 2a–d with different L_10_ fixed at 30 μm, 55 μm, 95 μm, and 140 μm, respectively. The ERs of the optical interference fringes are greatly diverse by varying the L_50_. We can analyze the interference fringes and demonstrate that the FSR is individually dominated by the L_10_ of HCF_10_, which generates the interference spectra. However, the length L_50_ of HCF_50_ acts as a fiber beam splitter for different beam expansions that greatly modify the ER of interference spectra. 

To obtain the optimal interference for the I_co_, which is almost the same as the I_cl_, we further improve the resolution of the length L_50_, with an interval of 0.2 μm being one of the values of L_50_ that can obtain the best ER of interference spectra over the wavelength range covered by the S + C + L optical fiber communication bands, as shown in Figure 3. 

Figure 3 shows the optimal ERs of the optimal interferences that can be obtained at each case of L_10_ fixed at 30 μm, 55 μm, 95 μm, and 140 μm when the beam splitter HCF_50_ with L_50_ is 89.4 μm, 84.4 μm, 82.2 μm, and 70.8 μm, respectively. One can see that the ER over 50 dB (around λ = 1550 nm) can be almost achieved by arranging the lengths of the HCF_50_ and HCF_10_. 

From the above simulation results (Figure 2 and Figure 3), Figure 4 shows the FSRs of the optical interference in the proposed leaky-guided FMZI with varying lengths of HCF_50_ (L_50_) by fixing the HCF_10_ (L_10_). We can see that no matter the L_50_, it does not impact the FSR of the interference spectra. The results display the FSR is only affected by the L_10_ of the HCF_10_. Once the length of HCF_10_ is determined, the FSR is almost constant. Figure 4 shows the FSR values at the wavelength dips close to λ = 1550 nm. The results demonstrate that the FSRs are 149.9 nm, 89.8 nm, 49.7 nm, and 34.5 nm when the L_10_ are 30 μm, 55 μm, 95 μm, and 140 μm, respectively. Therefore, one can adjust the length of L_10_ to gain the desired FSR in the interference spectra.

In addition, Figure 5 indicates the ERs of the proposed leaky-guided FMZI with varying L_50_ of HCF_50_ when the L_10_ of HCF_10_ is fixed at 30 μm, 55 μm, 95 μm, and 140 μm, respectively. From the results, we can observe the conditions of the highest ER of the interferences (the peaks in Figure 5) that occur in every specific L_50_ at each L_10_. The collocation of optimal interference conditions occurring in longer L_10_ would be arranged with a shorter L_50_, and vice versa. Here, we also can see the proposed structure with a longer L_50_ over 120 μm cannot have useful interference ERs. This is because a long L_50_ spreads the optical light more into the cladding of HCF_10_, which greatly reduces light intensity in the core of HCF_10_ to make the difference between the I_co_ and I_cl_ even greater. Additionally, the optical light in the leaky core of HCF_10_ gradually leaks out into the silica cladding through the increasing propagation distance. Thus, the much greater difference between the I_co_ and I_cl_ results in a poor ER. It is obvious that to fabricate such a high-quality interferometer, the length required accuracy of each section in the HCF’s connection configuration is exceptionally strict. 

To verify the optimal structure proposed by theoretical simulation, we perform the experiments. Our process begins with the crucial step of cleaving the SMF, which has an outer diameter of 125 μm and an inner diameter of 8.2 μm. This is followed by the fusion splicing with HCF_50_ and the cleaving of an appropriate length (L_50_) to approach the optimal L_50_. Another segment based on the HCF_10_ follows the same steps. Finally, the two segments, the SMF-HCF_50_ and HCF_10_-SMF, are aligned with their optimal lengths (L_50_ and L_10_) and then arc discharge of fusion splicing. The fabrication flow chart is displayed in Figure 6a. This method ensures precise fiber connections and provides a reliable foundation for subsequent experiments. Figure 6b illustrates the experimental setup for spectral measurement. A broadband light source (BLS) with wavelengths of 1250~1650 nm is incident from input-SMF to the HCF_50_-HCF_10_, inducing the cladding mode and interfering with the leaky-guided core mode in output-SMF. The reliability of the optical spectrum analyzer (OSA) in measuring the interference spectra ensures the accuracy of the results.

Figure 7 presents the theoretical simulation and experimental results for the different combinations of the L_50_ and L_10_ for the optimized spectral interference. Their optical interference spectra are compared and analyzed. The red solid lines represent the experimental data and the highest ER over 40 dB achieved, while the blue dashed lines represent the simulation results of the optimal condition. The insets show their corresponding microscope photographs of the fabricated FMZIs. The values of (a) 90.5/28.8 μm, (b) 86.7/55.9 μm, (c) 81.5/96.6 μm, and (d) 72.5/140 μm are measured by the optical microscope with an error of ±1 μm. We can see that the experimental and simulation results are in good agreement in Figure 7b–d except Figure 7a. The fabricated structure of the lengths L_50_ and L_10_ demonstrates that the simulation work is helpful for the experiment approaching optimal performance. It is worth noting that there is a significant discrepancy between the theoretical value and the experimental value in Figure 7a with such a tiny L_10_. This is because the production accuracy of the ultrashort L_10_ is more difficult to achieve. The required accuracy of the lengths of L_50_ as well as the L_10_ in each section of the configuration is especially rigorous. Thus, using the commercial fiber cleaver and general fusion splicer to fabricate the proposed HCF-based fiber device extremely precisely in the experiment is not easy. However, the developed simulation work for the refined fiber interferometers is significantly valuable for the design, fabrication, and operation. In practical applications of the proposed LGFW-FMZI, we can control the FSR by altering the length of HCF_10_. Once the FSR is determined, the length of HCF_50_ can be selected to satisfy the requirements for the designed fiber optic interferometer. This makes the designed devices more flexible and practical. Therefore, once the optimally designed structure is obtained, the proposed ultrahigh extinction ratio fiber device can be designed for some fiber passive devices in fiber communications and fiber laser technology applications, like a band-pass filter and a wavelength-selective filter. Moreover, creating a microhole in the HCF_50_ section is accessible using an fs laser. One can introduce different materials into the HCF [6]. For example, the specimens of chemical and biological detection. Especially for measuring some precious and rare materials since the volume required is merely picoliter level. Due to the high fringe visibility that can achieve a high-resolution measurement, the proposed approach can be applied to measure materials with high accuracy, demonstrating its adaptability to various measurements.

It is particularly mentioned that silica fiber’s thermal expansion coefficient (a) is low, about 5 × 10^−7^ (1/K), its Young’s modulus is very high, approximately 70 GPa, and inelastic in such a tiny HCF of tens micrometers. Therefore, it is insensitive to thermal, bending, and strain parameters when non-filling material in the hollow core. Such features would benefit from avoiding cross-sensitivity in sensing technology.

## 4. Conclusions

We have proposed a novel, ultrahigh ER and ultracompact leaky-guided FMZI based on a tiny HCF_10_ front connected with a large core of a 50 μm HCF_50_ beam splitter. The main interference region in the proposed leaky-guided FMZI is the HCF_10_ section. The operated HCF_50_ expands the optical beam and splits into two paths, launching into the HCF_10_ section. It assists the interference section of the HCF_10_ to improve the optimal interference (ultrahigh ER). By varying the micro-lengths of the HCF_50_ (L_50_) and HCF_10_ (L_10_) sections, we first demonstrated the light splitting and combining mechanisms on the designed FMZI. Simulation and experimental results indicate that the L_10_ of the HCF_10_ primarily influences the FSRs of the interference transmission spectra. However, the ERs would be affected by the L_50_ of beam splitter HCF_50_ when the L_10_ of HCF_10_ is fixed. Based on the simulation results, the greatest performance of ultrahigh ER, generally over 50 dB, can be achieved by the structures of combination: L_50_/L_10_ of 89.4 μm/30 μm, 84.4 μm/55 μm, 82.2 μm/95 μm, and 70.8 μm/140 μm. For each designed FMZI with a fixed L_10_, there will be a specific L_50_ to be matched to generate the optimal interference. However, the required accuracy of the lengths of L_50_ as well as the L_10_ in each section of the configuration is very exceptionally strict. Even so, the experimental and simulation results are still in acceptable agreement to demonstrate that the work is still valuable for the experiment approaching optimal performance. Finally, by appropriately varying the lengths L_50_ of the beam splitter and L_10_ of the interference length, the excellent interference spectrum characteristics with an ultrahigh ER and arbitrary FSR of the optical interference spectra in the proposed ultracompact LGFW-FMZI have been proposed. We believe the proposed fiber-optic device would benefit sensing and fiber communications technology applications.

## Figures and Tables

**Figure 1 nanomaterials-14-01494-f001:**
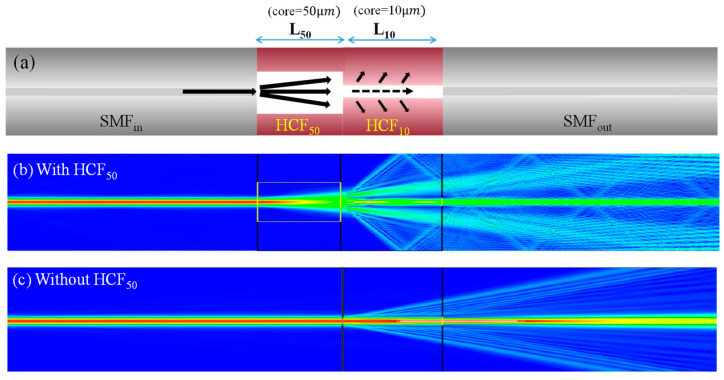
(**a**) Schematic diagram of the proposed HCF-based LGFW-FMZI. The FDBPM simulation results of optical field distribution of light for the proposed structure (**b**) with and (**c**) without the segment of the HCF_50_ beam splitter.

**Figure 2 nanomaterials-14-01494-f002:**
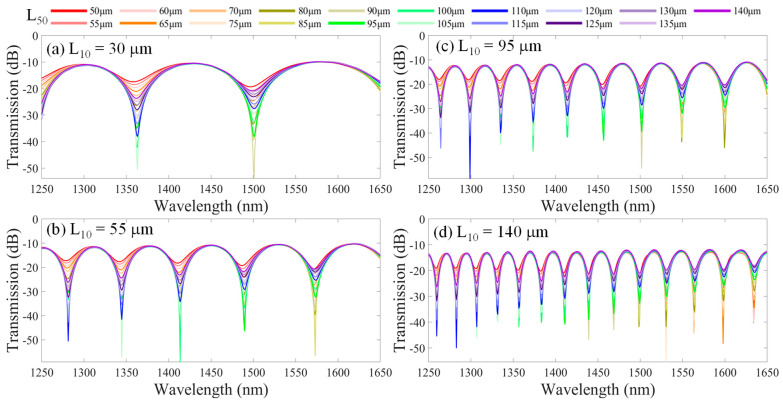
Simulated interference spectra for different combinations of L_50_ and L_10_ in the proposed leaky-guided FMZI with varying L_50_ from 50 to 140 μm and fixed L_10_ are (**a**) 30 μm, (**b**) 55 μm, (**c**) 95 μm, and (**d**) 140 μm, respectively.

**Figure 3 nanomaterials-14-01494-f003:**
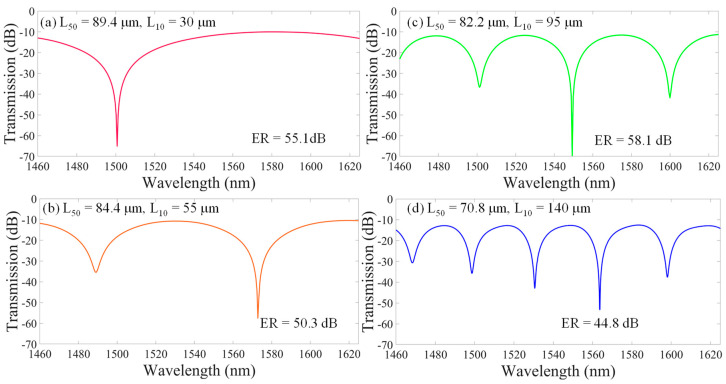
Simulation results for the best ultrahigh ER around 50 dB of the optimal interferences at L_10_ fixed at (**a**) 30 μm, (**b**) 55 μm, (**c**) 95 μm, and (**d**) 140 μm when the L_50_ is 89.4 μm, 84.4 μm, 82.2 μm, and 70.8 μm, respectively. (Note: the monitored wavelength range is 1460 nm–1625 nm).

**Figure 4 nanomaterials-14-01494-f004:**
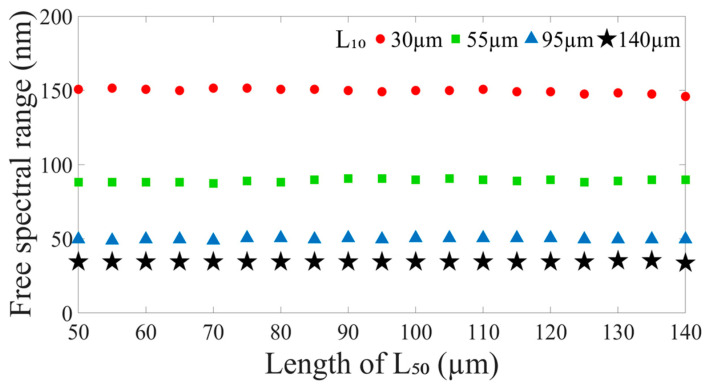
The FSR of the proposed leaky-guided FMZI with varying L_50_ of HCF_50_ when the L_10_ of HCF_10_ is fixed at (
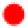
) 30 μm, (
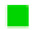
) 55 μm, (
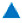
) 95 μm, and (
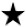
) 140 μm, respectively.

**Figure 5 nanomaterials-14-01494-f005:**
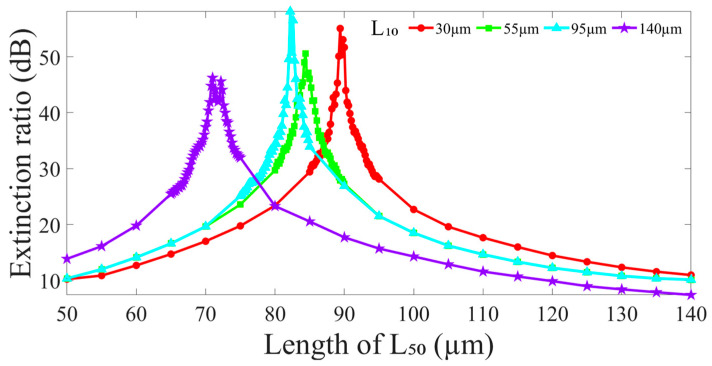
The ERs of interference fringe around λ = 1550 nm in the proposed leaky-guided FMZI with varying L_50_ of HCF_50_ when the L_10_ of HCF_10_ is fixed.

**Figure 6 nanomaterials-14-01494-f006:**
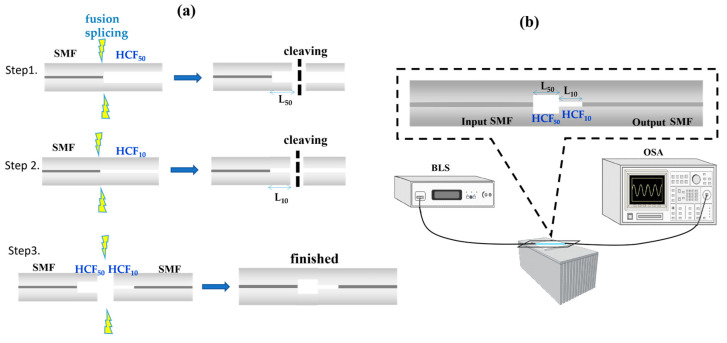
Schematic diagrams of (**a**) the fabrication steps and (**b**) the measurement system.

**Figure 7 nanomaterials-14-01494-f007:**
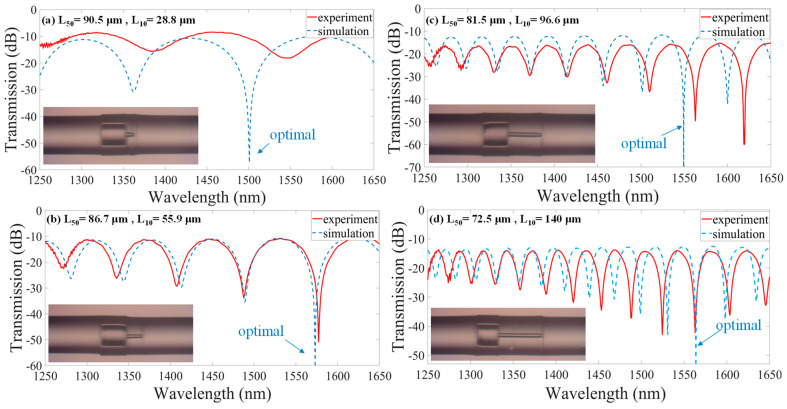
Comparison of the optical interference spectra of the simulation (optimal) and experimental results with fabricated L_50_/L_10_ (**a**) 90.5/28.8 μm, (**b**) 86.7/55.9 μm, (**c**) 81.5/96.6 μm, and (**d**) 72.5/140 μm, respectively. Their insets include a corresponding microscopic image of the fabricated devices.

## Data Availability

Data are contained within the article.

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
