# Peer review of "Ultrahigh Extinction Ratio Leaky-Guided Hollow Core Fiber Mach–Zehnder Interferometer Assisted by a Large Core Hollow Fiber Beam Splitter"

_nanomaterials, 2024, doi:10.3390/nano14181494_

Round 1

Reviewer 1 Report

Comments and Suggestions for Authors

The manuscript presents a modal interferometer in optical fiber based on two sections of hollow fibers inserted between the single mode feeding and output fibers. The firs hollow fiber of larger diameter controls the visibility of the interferometric pattern while the length of the thinnest diameter fiber (interferometric fiber) controls the free spectral range.

The manuscript is well organized and written, placed in a proper context and the basic ideas clearly exposed. Although the experimental results are similar to others previously reported in literature (see for example DOI 10.1109/JPHOT.2017.2671437) this manuscript presents an alternative approach.

  •  

Reviewer 2 Report

Comments and Suggestions for Authors

For comments see the attached file.

Reviewer 3 Report

Comments and Suggestions for Authors

This work proposes and demonstrates an interferometric optical fiber structure based on an air cavity. The authors splice two different hollow fibers to generate an in-line interferometer. Moreover, the authors present a compressive theoretical validation. These interferometric optical fiber structures have been widely proposed and demonstrated. Moreover, the interferometric response is evident, and any new surprise is repotted or presented in the manuscript. Furthermore, the authors do not demonstrate any potential application. In the present form, the manuscript is not suitable for publication  in Nanomaterials,

*It is hard to provide a general discussion about the work. The authors use an excellent simulation model, demonstrate an interferometric optical fiber structure, and discuss the ER as a design parameter. However, the work is incomplete and has not shown potential applications such as fiber optic sensors or wavelength-selective filters for fiber lasers. According to the status of interferometric fiber optic devices, the fabrication process is not a merit of publication.

*The work would benefit from comprehensively validating the thermal, bending, and strain sensitivities. If the potential application of these devices is indeed a sensor, these sensitivities are crucial and should be fully validated to ensure the reliability and applicability of the proposed structure.

*The fabrication process, a crucial aspect of this work, should be clearly described and illustrated using a suitable sketch. This will greatly aid the reader in understanding the methodology and novelty of the work and ensure that the work is fully comprehensible and transparent.

*The experimental setup for interferometric setup characterization is a critical component that must be included and thoroughly discussed. This information will help the reader evaluate the reliability of the results.

*Recently, the authors proposed similar structures

https://doi.org/10.3390/polym14224966

https://doi.org/10.3390/s22030808

What is the contribution of the submitted work?

Round 2

Reviewer 2 Report

Comments and Suggestions for Authors

The revised version of the paper was significantly improved and all my comments, questions, and recommendations were answered appropriately, therefore I recommend accepting the paper for publication without additional comments and recommendations.

Author Response

We greatly appreciate the reviewer for the positive comments.

Reviewer 3 Report

Comments and Suggestions for Authors

In this new version, some concerns were addressed. However, the authors  must consider the following comments:

*It is hard to provide a general discussion about the work. The authors use an excellent simulation model, demonstrate an interferometric optical fiber structure, and discuss the ER as a design parameter. However, the work needs to be completed, and potential applications such as fiber optic sensors or wavelength-selective filters for fiber lasers have yet to be shown. According to the status of interferometric fiber optic devices, the fabrication process is not a merit of publication.

Response: Many thanks for the reviewer's comments. The main contribution of the study is to develop a new concept of the optical hollow-core fiber Mach–Zehnder interferometer assisted by a large core hollow fiber (HCF) beam splitter. The structure is based on two sections of HCF inserted between the single-mode feeding and output fibers. The first HCF50 of larger diameter as a beam splitter dominates the interferometric pattern's fringe visibility (extinction ratio, ER), and the length of the small diameter HCF10 controls the free spectral range. A comprehensive theoretical model is accomplished for performing the simulation calculations to optimize the  2 proposed structure to achieve the best interference extinction ratio with a desired FSR. By precisely adjusting the lengths of HCF10 and HCF50, the proposed fiber interferometers can perform the capability of an ultrahigh ER over 50 dB with the arbitrary FSR in the transmitted interference spectra over an ultra-broad wavelength range of 1250nm to 1650nm. These performances make the designed devices more flexible and practical. In practical applications, the proposed ultrahigh extinction ratio fiber device can be designed for fiber-optics passive devices in fiber communications or fiber laser technology once the optimal design is obtained. They can be applied as a band-pass filter or a wavelength-selective filter for practical applications. Moreover, drilling a microhole in the HCF50 section is easy using an fs-laser. One can introduce different materials into the HCF via the microhole (please refer to our previous work [6] and the following figure). The filling materials, such as chemical and biological detection specimens, can be diverse. It is beneficial as a fiber sensor for measuring precious and rare materials since the volume required for detection is merely at the picoliter level. Due to the high fringe visibility that can achieve a high-resolution measurement, the proposed approach can be applied to measure materials with high accuracy. Although we have yet to apply the proposed fiber device to the application of a practical system, the theoretical model, numerical simulations, characteristic analysis, and experiment of the designed devices have been comprehensively accomplished. A brief description of the possible applications has been added in section 3 (rows 276- 286) of the revised manuscript. Micrograph of the drilled microhole on the HCF surface by fs-laser [6].

Reply:

Although the authors include new information, and the present version is interesting, as previously mentioned, the fabrication and characterization do not contribute to the literature related to interferometric optical fiber sensors. Moreover, the thermal, bending, and strain responses should be included. Here, the authors must demonstrate a potential application. The manuscript will be suitable for publication once the authors attend to this comment.
